# Process Optimization, Structural Characterization, and Calcium Release Rate Evaluation of Mung Bean Peptides-Calcium Chelate

**DOI:** 10.3390/foods12051058

**Published:** 2023-03-02

**Authors:** Wenliang Zhai, Dong Lin, Ruoshuang Mo, Xiaozhuan Zou, Yongqing Zhang, Liyun Zhang, Yonghui Ge

**Affiliations:** 1College of Food and Pharmacy Engineering, Guiyang University, Guiyang 550005, China; 2Key Laboratory of Functional Food of Universities in Guizhou Province, Guiyang 550005, China; 3Biopharmaceutical Engineering Research Center of Guizhou Province, Guiyang 550005, China

**Keywords:** mung bean, peptides-calcium chelate, calcium supplement, structural characterization, simulated digestion

## Abstract

To reduce grievous ecological environment pollution and protein resource waste during mung bean starch production, mung bean peptides-calcium chelate (MBP-Ca) was synthesized as a novel and efficient calcium supplement. Under the optimal conditions (pH = 6, temperature = 45 °C, mass ratio of mung bean peptides (MBP)/CaCl_2_ = 4:1, MBP concentration = 20 mg/mL, time = 60 min), the obtained MBP-Ca achieved a calcium chelating rate of 86.26%. MBP-Ca, different from MBP, was a new compound rich in glutamic acid (32.74%) and aspartic acid (15.10%). Calcium ions could bind to MBP mainly through carboxyl oxygen, carbonyl oxygen, and amino nitrogen atoms to form MBP-Ca. Calcium ions-induced intra- and intermolecular interactions caused the folding and aggregation of MBP. After the chelation reaction between calcium ions and MBP, the percentage of β-sheet in the secondary structure of MBP increased by 1.90%, the size of the peptides increased by 124.42 nm, and the dense and smooth surface structure of MBP was transformed into fragmented and coarse blocks. Under different temperatures, pH, and gastrointestinal simulated digestion conditions, MBP-Ca exhibited an increased calcium release rate compared with the conventional calcium supplement CaCl_2_. Overall, MBP-Ca showed promise as an alternative dietary calcium supplement with good calcium absorption and bioavailability.

## 1. Introduction

Calcium is one of the essential macroelements abundant in the human body, accounting for approximately 1.5–2% of the body weight, and 99% of calcium is found in human bones and teeth [1]. Calcium is not only importantly involved in the basic physiological performance of humans, but also widely implicated in multiple physiological processes such as muscle contraction, blood coagulation, nerve conduction, and enzyme catalytic activity [2]. Inadequate calcium intake can lead to multiple diseases such as rickets, osteoporosis, colon cancer, and hypertension [3,4,5]. Therefore, an adequate daily calcium intake is essential for the human body to maintain health. At present, inadequate calcium intake is prevalent all over the world, and particularly, the calcium intake is generally less than 700 mg/day in the populations of developing countries in Asia, Africa, Latin America, etc, which is lower than recommended daily allowance (RDA) ranges (700 to 1200 mg) [6]. Calcium cannot be produced through the metabolism of the human body itself and its intake can be only achieved through the daily diet. However, calcium in the daily diet is mainly present in ionic form in the digestive tract and is very likely converted to Ca(OH)_2_ precipitate in the alkaline environment of the intestine, thus seriously affecting the absorption of calcium [7]. Therefore, the daily intake of calcium supplements as a dietary supplement is extremely important for the maintenance of human health. Several main calcium supplements have been extensively applied in the current market, including inorganic calcium (such as CaCO_3_ and CaCl_2_), organic calcium (such as calcium gluconate and calcium lactate), and amino acid calcium (such as calcium glycinate and calcium aspartate) [8,9]. However, inorganic calcium is prone to be transformed into insoluble precipitate in the intestine and its bioavailability is extremely low. Additionally, the intake of inorganic calcium is also often accompanied by adverse effects such as constipation and flatulence [10]. Although organic calcium partly increases the bioavailability of calcium, it has significant antagonistic effects on the absorption of other nutrients. In contrast to inorganic and organic calcium, amino acid calcium can effectively increase the release rate of calcium ions in the gastrointestinal tract [11]. However, high expense, pro-oxidant effect on fats, and undesirable color reaction severely restrict its application as a calcium supplement. Hence, it seems to be extremely urgent to develop novel calcium supplements that effectively overcome the disadvantages mentioned above.

Currently, increasing researchers have paid attention to the utility of peptides-calcium chelates as a potential dietary calcium supplement; animal-derived and plant-derived peptides-calcium chelates including black bean [12], walnut [13], chicken foot [14], and tilapia scale [15], are continuously being reported. Peptides-calcium chelates can be more easily absorbed and have better bioavailability than conventional calcium supplements [8]. Wu et al. [16] found that the calcium absorption rate of octopus scraps protein hydrolysate-Ca chelate increased by 41% compared to calcium supplement CaCl_2_ in Caco-2 cells. According to Sun et al. [17], Herring egg phosphopeptides-Ca supplementation significantly enhanced calcium absorption and serum calcium levels in calcium-deficient mice. The oxygen and nitrogen atoms on carbonyl, carboxyl, and amino groups in peptides have lone pair electrons to pair with electron-deficient Ca^2+^ through the coordination bond, thus forming chelates with a unique structure. Through the transfer of electrons, the calcium in the chelate becomes electrically neutral and more stable. Thus, during the digestion process, the protection of peptides prevents the calcium precipitation caused by the alkaline environment of the intestine or the interaction with the food matrix, thereby promoting the efficient release of calcium and increasing its bioavailability. Peptides-calcium chelates deserve more attention and research owing to the excellent bioavailability of calcium, and more efforts should be made to focus on more calcium-chelating bioactive peptides.

Mung bean (*Vigna radiata* L.) is widely grown in Asia and other countries worldwide, covering more than 6 million hectares, accounting for approximately 8.5% of the global legume planting land [18]. Mung bean is rich in nutrients such as starch, vitamins, minerals, proteins, and essential amino acids, of which mung bean protein is the second most abundant component (25–28%) secondary to starch and is also a high-quality protein with amino acid composition characteristics that fully meet the criteria recommended by FAO/WHO [19]. In many developing countries, such as China and India, mung bean protein is extensively applied as an alternative to animal protein in product formulations [20]. However, as a raw material in the food industry, mung beans are mainly used for starch production rather than protein production. A large amount of protein-rich industrial wastewater (protein concentration 3.1%, *w/v*) is discharged during mung bean starch production [21], which not only causes a grievous waste of protein resources but also pollutes the ecological environment. A series of recent studies have shown that MBP obtained by hydrolyzing mung bean protein enzymatically have diverse physiological functions, such as anti-oxidant [22], anti-hypertensive [20], anti-tumor [23], and immune-enhancing effects [18], indicating that MBP is a potential source of functional components. Currently, the potential of MBP-Ca as calcium supplements has rarely been reported. Using techniques for ultrafiltration and peptide sequence analyses, Budseekoad et al. [24] identified three pentapeptides (LLLGI, AIVIL, and HADAD) with good calcium-ion binding capacity in mung bean protein hydrolysates. However, little is currently known about the preparation process and structural features of MBP-Ca as well as its calcium release rate under different conditions. Of note, understanding the information above is essential for pre-launch clinical trials, safety evaluation, and industrial production of MBP-Ca. Therefore, the study was guided by the following three objectives: (1) To determine the optimal preparation conditions of MBP-Ca through orthogonal design; (2) To characterize the structure and composition of MBP-Ca; (3) To measure the calcium release rate of MBP-Ca under varying temperatures, pH, and simulated gastrointestinal digestion conditions.

## 2. Material and Methods

### 2.1. Material and Chemicals

MBP (≥90.0% in mass fraction purity) was procured from Zhongshi Duqing Biotechnology Co., Ltd. (Heze, China), and CaCl_2_ (≥95% in mass fraction purity) was purchased from Sinopharm Chemical Reagent Co., Ltd. (Shanghai, China). Unless otherwise stated, analytical grade chemical reagents were used in this study.

### 2.2. Preparation of MBP-Ca

MBP-Ca was prepared using a modified method as described in a prior study [9]. MBP was dissolved in deionized water (≤1.3 μS/cm in conductivity) at various peptide concentrations (10, 20, 30, 40, and 50 mg/mL) and subsequently added with CaCl_2_ at different mass ratios (2:1, 4:1, 6:1, 8:1, and 10:1). The obtained mixture was reacted in a water bath for different periods (40, 50, 60, 70, and 80 min), at different temperatures (25, 35, 45, 55, and 65 °C), and different pH values (5, 6, 7, 8, and 9). 10 mL reaction solution was aspirated for subsequent calcium chelating rate determination. Thereafter, absolute ethanol (9× volume) was added to the product solution for the removal of the unreacted free Ca^2+^ and MBP. After 20 min, the mixture was centrifuged at 4000 r/min for another 20 min, and the precipitate was collected and lyophilized at −60 °C for 24 h. Ultimately, the lyophilized material was designated as MBP-Ca.

With the calcium chelating rate as the response value, four-factor and three-level orthogonal tests were designed to determine the optimal preparation conditions of MBP-Ca based on the results of single-factor tests. The calcium chelating rate was calculated using ethylenediamine tetraacetic acid (EDTA) complexometric titration reported by Wang et al. [25]. Half of the 10 mL reaction solution was titrated with EDTA as described in the literature [25], and the volume of EDTA consumed here was recorded as V_1_. Subsequently, absolute ethanol (9× volume) was added to the remaining reaction solution (5 mL) to obtain calcium in the chelating form. Thereafter, the titration test was performed and the volume of EDTA consumed here was recorded as V_2_. The calcium chelating rate was calculated according to the following equation:(1)$$\mathrm{Calcium}\;\mathrm{chelating}\;\mathrm{rate}\;\left(\%\right)=\frac{V2}{\;V1}\text{×}100\%$$
where V_1_ and V_2_ represent the volumes of total and chelating calcium, respectively.

### 2.3. Structural Characterization of MBP-Ca

#### 2.3.1. Amino Acid Composition Analysis

The amino acid composition of the samples was determined using an automated amino acid analyzer (L-8900, Hitachi Co., Tokyo, Japan) based on the previously reported method [26]. In detail, 5 mg MBP or MBP-Ca was transferred to a hydrolysis tube, which was mixed with 10 mL 6 mol/L HCl. After air discharge through N_2_ inflation, the hydrolysis tube was sealed and subsequently hydrolyzed at 110 °C for 24 h. After cooling, the hydrolysate solution was filtered through the 0.45 μM filter membrane, followed by amino acid composition analysis.

#### 2.3.2. Fluorescence Spectroscopy

Calcium ion-induced peptide conformational alterations were measured using an F-280 fluorescence spectrometer (Gangdong Co., Tianjin, China). At the excitation wavelength of 280 nm and emission wavelength range of 300–450 nm, fluorescence intensity of the samples was measured, with a scan speed of 240 nm/min, the excitation and emission slit widths of 10 nm, and the photomultiplier tube (PMT) voltage of 450 V. MBP-Ca was prepared as follows: CaCl_2_ (25, 50, 100, 200, 300, 400 mg/mL) was added in 0.05 mg/mL MBP solution and incubated at room temperature for 30 min.

#### 2.3.3. Fourier Transform Infrared (FTIR) Spectroscopy

The FTIR spectra of samples were determined using a modified method based on the previous study [27]. Briefly, 2 mg MBP or MBP-Ca was well mixed with 200 mg dried KBr, ground to a fine powder, and compressed into a transparent sheet. Then, the sheet was loaded on the FTIR instrument (iS 50, Thermo-Nicolet Co., Madison, WI, USA) for infrared spectroscopic analysis. The spectra from 4000 to 400 cm^−1^ were recorded using an FTIR spectrometer with a resolution of 4 cm^−1^ and a scan number of 64.

#### 2.3.4. Circular Dichroism (CD) Spectroscopy 

The secondary structures of MBP and MBP-Ca were determined using a Jasco J-1500 CD spectroscope (JASCO Co., Tokyo, Japan) based on a previously reported method [28]. With the distilled water as a reference, each sample (0.1 mg/mL) was scanned three times using a quartz cell with a pathlength of 0.1 cm and a scan speed of 100 nm/min, and the scanned spectral data within the range of 190–260 nm were recorded.

#### 2.3.5. Scanning Electron Microscopy (SEM) and Energy Dispersive X-ray Spectrometer (EDX) Analysis

The microstructures of MBP and MBP-Ca were observed under SEM (JSM-5510, JEOL Co., Tokyo, Japan) as described in the literature [29]. Before imaging, the lyophilized sample powder was smeared on a glass slide, which was fixed to the sample holder using double-sided adhesive tape, and the sample was sprayed with thin-layer gold film. At an accelerating voltage of 10 kV, the micrographs of samples (5000 times) were obtained. The surface elemental compositions of MBP and MBP-Ca were analyzed using EDX (EX-250, Horiba Ltd., Tokyo, Japan), for which the samples were prepared using the sample preparation method in SEM.

#### 2.3.6. Particle Size Distribution Analysis

The particle size distribution of MBP and MBP-Ca was determined using a laser particle size analyzer (Zetasizer Nano-ZS90, Malvern Instruments Ltd., Malvern, Worcestershire, UK) based on the method as previously described [29]. Before the test, the samples were formulated into 1 mg/mL solution using distilled water. Afterwards, a polystyrene pool (1 cm) was filled with 1.5 mL of the sample and allowed to stand at 37 °C for two minutes before measurement.

### 2.4. Determination of Calcium Release Rate

To investigate the stability of MBP-Ca during processing and storage, calcium release rates of MBP-Ca were determined at different pH and temperature conditions using CaCl_2_ as a control. An aqueous solution containing 5 mg/mL of MBP-Ca or CaCl_2_ was prepared, which was modified to different pH values (1, 3, 5, 7, 9, and 11) and then incubated at 37 °C for 1 h. For the temperature test, 5 mg/mL MBP-Ca and CaCl_2_ solution (pH = 7) were incubated for 1 h at 25 °C, 40 °C, 60 °C, 80 °C, and 100 °C. All samples mentioned above were centrifuged at 4000 r/min for 20 min at room temperature to remove insoluble calcium [30]. Calcium content was determined using an atomic absorption spectrophotometer (AA-7000, SHIMADZU Co., Kyoto, Japan). The operating parameters were as follows: wavelength of 422.7 nm, slit of 0.5 nm, current of 10 mA, air flow of 10 L/min, and acetylene flow of 2 L/min. The calcium release rate was calculated according to the following formula:(2)$$\mathrm{Calcium}\;\mathrm{release}\;\mathrm{rate}\;(\%)=\frac{\mathrm{Calcium}\;\mathrm{content}\;\mathrm{in}\;\mathrm{supematant}}{\mathrm{The}\;\mathrm{total}\;\mathrm{calcium}\;\mathrm{content}\;}\text{×}100\%$$

In vitro simulated gastrointestinal digestion tests were implemented using a modified method reported by Fu et al. [31]. After 0.2 g NaCl and 0.32 g pepsin were dispersed into 100 mL distilled water, the pH value was adjusted to 2 to obtain simulated gastric fluid. The simulated intestinal fluid was prepared as follows: 1 g trypsin was added to 100 mL distilled water and adjusted to the pH value of 7.4 using 0.2 mol/L NaOH. At first, 5 mg/mL MBP-Ca solution was preheated in a 37 °C water bath for 10 min, followed by the addition of simulated gastric fluid based on the enzyme-to-substrate ratio of 1:50 *(w/w)*. After the simulated gastric digestion lasted for 120 min, the pH value was raised to 7.4, and the simulated intestinal digestion was initiated with the addition of the simulated intestinal fluid according to the enzyme-to-substrate ratio of 1:50 *(w/w).* At the time points of 0, 30, 60, 90, 120, 150, 180, 210, and 240 min, respectively, the test samples were harvested and immersed in boiling water at 100 °C for 10 min to inactivate the enzyme. After cooling, the samples were centrifuged at 4000 r/min for 20 min, and the supernatant was collected, in which the calcium content and calcium release rate were calculated in the same way as mentioned above. CaCl_2_, a conventional calcium supplement, was used as a positive control.

### 2.5. Statistical Analysis

All determinations were performed in triplicate, and all data were expressed as mean ± standard deviation of the mean. Differences between the means were analyzed by Student’s *t*-test for two groups, and by one-way analysis of variance (ANOVA) followed by Duncan’s multiple range test for multiple comparisons. The means were considered significantly different at *p* < 0.05. SPSS (version 18.0, SPSS Inc., Chicago, IL, USA) software was used for statistical analysis.

## 3. Results and Discussion

### 3.1. Optimization of the Preparation Conditions of MBP-Ca

Varying processing conditions may have different impacts on the calcium chelation capacity. To improve the preparation efficiency of MBP-Ca, this study assessed the effects of temperature (°C), time (min), mass ratio of MBP/CaCl_2_ (*w*/*w*), pH, and MBP concentration (mg/mL) on the calcium chelating rate. As shown in Figure 1A, with the increment of the chelating time from 40 to 60 min, the calcium chelating rate increased rapidly from 79.22% to 83.02%, reaching the maximum value, while further prolongation of the reaction time seemed to have little effect on the calcium chelating rate. Since MBP can react with calcium ions within a relatively short time, the chelation reaction time was fixed to 60 min for the subsequent orthogonal test design to save time and economic costs. As shown in Figure 1B, as the pH value increased from 5 to 9, the calcium chelating rate first increased and then decreased, which reached a maximum value of 85.12% when pH = 7. When pH < 7, the increase of pH value would cause the decrease of H^+^ concentration and improve the electron-donating ability of potential metal-ion-chelating sites (such as -COOH and -NH_2_ in MBP), thus enhancing the binding ability of MBP to Ca^2+^. When pH > 7, OH^−^ with a high concentration would preferentially form Ca(OH)_2_ precipitate with Ca^2+^, which impaired the chelating reaction between MBP and Ca^2+^. Therefore, under a neutral environment, calcium ions were more prone to incorporate with the ligand groups of MBP, which was agree with the findings of Luo et al. [32]. The calcium chelating rate of MBP-Ca significantly increased from 76.60% to a peak value of 83.11% when the temperature increased from 25 °C to 45 °C (Figure 1C). Since the binding of peptides to metal ions is an endothermic reaction, a moderate increase in temperature contributes to accelerating the motion of the molecules, enhancing the collision chance between the molecules, and thus improving the reaction efficiency. Then, when the temperature increased to 65 °C, the calcium chelating rate significantly decreased to 80.05% (Figure 1C), which might be attributed to excessive temperature-caused partial or complete denaturation of MBP. The results of the present study were similar to the findings reported by Wu et al. [9]. It was found that with the increase of reaction temperature, the binding ability of porcine bone collagen peptides to calcium increased gradually, which reached the maximum at 50 °C, while the calcium-binding ability decreased with a further increase of the temperature [9]. A similar increase and subsequent decrease also occurred in the experiment that assessed the effect of the change in the MBP/CaCl_2_ mass ratio on the calcium chelating rate. While increasing the mass ratio from 2:1 to 4:1, the chelating ratio increased sharply from 60.72% to 84.17%. Interestingly, no significant change was observed when the chelating ratio was raised to the range of 4:1 to 6:1. When the peptide-to-calcium mass ratio was greater than 6:1, the chelating ratio showed a decreasing trend. It was illustrated that the MBP/CaCl_2_ mass ratio had a significant effect on the calcium chelating rate (Figure 1D). A low peptide-to-calcium mass ratio would lead to a low chelating rate, since the mass of peptides could not meet the requirements of chelation reaction, while a high MBP/CaCl_2_ mass ratio would reduce the efficient utilization of peptides, resulting in the waste of peptides and diseconomy. Similar results were also reported by Cui et al. that peptide-to-calcium mass ratios had similar changes during the preparation of sea cucumber ovum hydrolysate-calcium complex [33]. As shown in Figure 1E, the calcium chelating rate showed a pattern of first increasing and then decreasing in an MBP concentration-dependent manner, and it reached a peak value of 84.86% when the MBP concentration was equal to 20 mg/mL. This may be explained by a significantly increased risk of collision between the reactive groups of MBP and calcium ions when the concentration of peptides increased, contributing to the formation of MBP-Ca. However, too high MBP concentration could significantly reduce the mass fraction of moisture in the reaction system, which seriously hindered the diffusion and motion of MBP and thus inhibited the formation of chelates.

The orthogonal test design and results are outlined in Table 1. Based on the range analysis, we found that the effect of the single-factor process condition on the calcium chelating rate was ranked as C > A > D > B, namely, mass ratio of MBP/CaCl_2_ > pH > MBP concentration > temperature. The optimal preparation conditions of MBP-Ca (A_1_B_2_C_1_D_2_) contained a pH value of 6, a temperature of 45 °C, an MBP/CaCl_2_ mass ratio of 4:1, and an MBP concentration of 20 mg/mL. The calcium chelating rate reached 86.26% in the validation test under the above-mentioned optimal preparation conditions. The high calcium chelating rate in the chelates suggested that a successful optimization of MBP-Ca preparation conditions was achieved. The chelates prepared under these conditions could be used for subsequent structural characterization and calcium release rate evaluation.

### 3.2. The Structural Characteristics of MBP-Ca

#### 3.2.1. Amino Acid Composition

The amino acid compositions of peptides can directly affect their functional properties and biological activity [12]. An amino acid analyzer was employed to analyze differences in amino acid compositions between MBP and MBP-Ca, and the results are summarized in Table 2. It was reported that the content of carboxyl groups in soybean protein hydrolysates was linearly correlated with calcium ion binding capacity [34]. The carboxyl groups of Glu and Asp residues in peptides might be the main binding sites for calcium ions. As presented in Table 2, the proportion of negatively charged acidic amino acids in MBP increased from 33.45% to 47.84% after calcium ions chelation, and the relative contents of Asp and Glu increased from 12.14% and 21.31% to 15.10% and 32.74%, respectively. It was indicated that carboxyl (-COO^−^) groups on Asp and Glu residues in MBP might be the binding sites for chelation reaction. After coordination with Ca^2+^, the -COO^−^ on the side chain of acidic amino acids was transformed to -COOCa. Following ethanol precipitation, the acidic amino acids were enriched by calcium ions, finally increasing the proportion of acidic amino acids in MBP-Ca. Furthermore, it has been reported that basic amino acids also function in the metal chelation reaction and can potently strengthen the stability of chelates [35]. However, the relative content of basic amino acids in MBP-Ca did not increase after chelation with calcium ions but decreased from 16.89% to 15.54%. It may be due to the fact that both the basic amino acid side chains and Ca^2+^ are positively charged under the pH condition of MBP-Ca preparation. Strong electrostatic repulsion was generated between the two, which damaged the chelation reaction [36].

#### 3.2.2. Fluorescence Spectra

The phenylalanine (Phe), tyrosine (Tyr), and tryptophan (Trp) amino acid residues in proteins all have a π-conjugation system that can produce fluorescence, but their fluorescence spectra can be generated at different emission peak positions due to the subtle differences in the structures of the side-chain chromogenic groups [37]. As shown in Table 2, MBP contained Tyr and Phe amino acid residues, so we used fluorescence spectroscopy to reveal calcium ion-induced changes in the spatial structure of MBP. As depicted in Figure 2A, by virtue of the rising in CaCl_2_ concentration, the fluorescence absorption peaks at both 305.8 nm and 346.2 nm shifted to peaks at longer wavelengths 306 nm and 348.6 nm, respectively, corresponding to the characteristic fluorescence emission peak positions of Tyr and Trp residues in peptides [12]. Our results suggested that Tyr and Trp residues in MBP might bind to Ca^2+^ through π-cation interactions and change the spatial distribution of electrons in MBP, which affected the energy of the excited state and consequently altered the position of the fluorescence emission peaks [38]. Additionally, the intensities of the fluorescence absorption peaks at both sites increased gradually with the increase of CaCl_2_ concentration. A possible explanation is that synergistic interaction between Ca^2+^ and MBP can stepwise induce the transfer of chromophores entrapped inside MBP to the molecular surface [16]. These findings were similar to the results of Shao et al. [39] about the fluorescence spectral changes of the chelation reaction between sesame peptides and calcium ions. Collectively, after the MBP bound to calcium ions, both the position and the intensity of the emission peak of the fluorescence spectra were altered caused by the spatial structure changes, indicating MBP-Ca as a new substance was different from MBP.

#### 3.2.3. FTIR Spectra

FTIR spectrum has been widely used to confirm the chelating sites between active groups in peptides and metal ions [40,41]. The significant change in the position of the absorption peaks in FTIR spectra indicated that the organic ligand group in the peptides interacted with Ca^2+^. As depicted in Figure 2B, the infrared profiles of MBP-Ca and MBP showed significant differences. The vibrational spectra (1700–1500 cm^−1^) corresponded to the characteristic infrared absorption of amide I and amide II bands in amino groups. After binding to calcium ions, the peak of the amide I band (C = O stretching vibration) shifted from 1678.01 cm^−1^ to low-frequency 1608.58 cm^−1^ [38], and the peak of amide II band (C-N stretching vibration coupled to N-H deformation) at 1544.93 cm^−1^ disappeared [41], indicating that the chelation reaction between MBP and Ca ions involved both carbonyl and amino groups in peptide bonds. The frequency of the N-H stretching vibration in MBP was close to the first-order frequency doubling peak of the amide II band, and thus the Fermi resonance occurred, producing two absorption peaks: the amide A band at 3433.18 cm^−1^, and the amide B band at 3066.72 cm^−1^. After chelation with Ca^2+^, the position shifted toward the high-frequency direction at wavenumbers of 3502.62 and 3068.65 cm^−1,^ respectively. It might be attributed to the fact that the formation of a coordination bond between the lone pair electrons on the N atom in the N-H bond and Ca^2+^ led to the stretching of the N-H bond and changed the constant of the chemical bond, ultimately altering the infrared vibrational frequency [8]. Therefore, the N-H bond might be involved in the formation of MBP-Ca. The COO^−^ symmetric stretching vibration at 1452.35 cm^−1^ shifted to low-frequency 1384.84 cm^−1^ [42], indicating that COO^−^ might bind to Ca^2+^ through the covalent bond to form COO-Ca and thus be involved in the chelating reaction. The wavenumber of the amide III band (1388.70 cm^−1^) shifted to 1292.26 cm^−1^ after binding to calcium ions, indicating the alteration of C-N stretching vibration in MBP [43]. Upon binding to calcium ions, the wavenumber at 1076.24 cm^−1^ (C-O stretching vibration) shifted to high-frequency 1118.67 cm^−1^, suggesting the formation of a C-O-Ca bond. In addition, within the range of 500–900 cm^−1^, upon vibration of C-H and N-H bonds, several new absorption peaks occurred in the spectra of MBP-Ca as compared to MBP [38]. Based on the aforementioned results, we inferred that the binding sites between calcium ions and MBP were mainly located on the O and N atoms in the carboxyl, amino, and peptide bonds, which was similar to the finding of Zhou et al. regarding wheat germ protein hydrolysates-calcium complexes [44].

#### 3.2.4. CD Spectra

Intercalation of metal ions may lead to alterations of the secondary structure of peptides, and CD spectroscopy has been widely used in the study of changes in the secondary structure of proteins and peptides [8]. CD spectrum was applied to monitor the effect of Ca^2+^ on the secondary structure of MBP. As shown in Figure 2C, The CD spectrum of MBP showed a distinct negative peak around 197 nm, which was blue-shifted to about 195 nm after the formation of MBP-Ca. The secondary structure analysis showed that MBP was mainly composed of four secondary structures, including α-helix, β-sheet, β-turn, and random coil, in which random coil was the dominating structure, accounting for 41.40%. Upon the binding of calcium ions to MBP, the relative content of β-sheet increased by 1.90% (from 28.60% to 30.50%). Correspondingly, as compared to MBP, the contents of α-helix and random coil in MBP-Ca were reduced by 1.10% and 0.80%, respectively (Figure 2D). The formation of a covalent bond between Ca^2+^ and MBP induced the folding of the MBP spatial conformation, which resulted in the secondary structural transformation of α-helix and random coil toward more ordered and stable β-sheet, eventually leading to the changes in the morphology and size of MBP-Ca.

#### 3.2.5. Morphology and Elemental Compositions

SEM is commonly used for analyzing the surface microstructure of biomaterials [45]. Therefore, this method was adopted to analyze the morphological changes of MBP before and after chelation with Ca^2+^. As shown in Figure 3A,B, MBP-Ca had a significantly different microstructure from MBP. The MBP had a flat and smooth surface structure without fissures. In contrast, MBP-Ca was fragmented into isolated solid blocks with irregular shapes, showing a rough and uneven surface structure. The morphological changes suggested that during the formation of MBP-Ca, Ca^2+^ might interact with MBP, which destroyed the initial intact dense structure and led to the folding, opening, refolding, and aggregation of the peptides, and finally the formation of heterogeneous-sized blocks [46]. He et al. [47] revealed a similar phenomenon of tilapia bone peptide calcium chelate observed by SEM. This was also confirmed in the results of calcium ion-induced variations in the secondary structure of MBP: during the formation of complex, α-helix and random coil were gradually transformed into β-sheet (Figure 2C,D).

EDX is a widely used technique for chemical element characterization, which can elucidate the composition and distribution of elements on the surface of samples [48,49]. The surface elemental compositions of MBP-Ca and MBP were analyzed using EDX and the results are displayed in Figure 3C,D. MBP primarily consisted of three elements, namely, C (52.78%), N (18.00%), and O (23.87%); however, after reaction with calcium ions, two characteristic absorption peaks of Ca newly occurred in the EDX spectra when compared with the original peptides, and the main surface elemental composition was changed to four elements C (48.80%), N (12.90%), O (29.39%), and Ca (6.18%). The presence of the calcium element signal indicated that calcium ions successfully bound to MBP.

#### 3.2.6. Size Distribution

The size distribution, which correlates with the volume of the object, is an important physical parameter for the identification of nanoparticles, and the particle size can directly determine the scope of the sample application [50]. According to the representative particle size distribution of MBP-Ca and MBP nanoparticles (Figure 3E), the mean sizes of MBP and MBP-Ca were 224.25 ± 18.04 nm and 384.67 ± 9.92 nm, respectively. The scale of MBP-Ca was obviously bigger than that of MBP (*p* < 0.01), indicating that the intermolecular interaction between MBP and Ca^2+^ promoted the interconnections among different peptides [51], which increased the size of the complex particles. Additionally, the polydispersity index (PDI) values of MBP and MBP-Ca were both less than 0.4 (MBP = 0.327 ± 0.09; MBP-Ca = 0.297 ± 0.03). The results indicated that both MBP and MBP-Ca particles were highly concentrated and uniformly distributed. Similar to the results of this study, the particle size of Antarctic krill peptides was 309.73 ± 3.73 nm, and the particle size of the peptides-zinc chelate was increased to 3460.33 ± 146.36 nm after chelation [49]. Moreover, the results of particle size analysis had a good fitting with the results of SEM (Figure 3A,B), which validated that the reaction between MBP and Ca^2+^ could promote the folding and aggregation of MBP.

### 3.3. Calcium Release Rates of MBP-Ca

#### 3.3.1. Calcium Release Rates under Different pH Conditions

During human food intake, the food will pass through different pH environments such as the oral cavity, stomach, and intestine, and pH stability is essential for the efficient utilization of nutritional factors in food [52]. Figure 4A depicts the calcium release rates of CaCl_2_ and MBP-Ca under different pH conditions. The calcium release rates of both CaCl_2_ and MBP-Ca showed no significant difference within the pH range of 1–5 (*p* > 0.05), which remained stable at about 90%. Within the pH range of 7–11, although the calcium release rate of MBP-Ca decreased gradually in a pH-dependent manner, it was remarkably higher than that of CaCl_2_ (*p* < 0.01), showing a good calcium release ability. Under the condition of pH = 7 (close to the intestinal physiological pH value), the calcium release rate of MBP-Ca was still maintained at 87.34%; whereas, the calcium release rate of CaCl_2_ (65.99%) was statistically lower than that of MBP-Ca (*p* < 0.01). The above-mentioned results suggested that MBP-Ca had better tolerance to the pH condition than the conventional calcium supplement CaCl_2_ and it had a good solubility, especially under alkaline conditions, indicating that MBP-Ca might have the potential to enhance the absorption of calcium ions via preventing the precipitation in the human gastrointestinal tract [48]. This may be explained by the fact that the H^+^ content is relatively high under acidic conditions, which can compete with Ca^2+^ to bind to the electron-donating group (-NH_2_ or -COOH) in MBP, thus inducing the release of abundant calcium ions. Under alkaline conditions, Ca^2+^ in CaCl_2_ can form Ca(OH)_2_ precipitate via binding to OH^−^, resulting in a sharp decrease in the content of free calcium ions [53]. However, MBP could bind to Ca^2+^ through the coordination bond and lead to the folding of the spatial structure of peptides, which enclosed Ca^2+^ to protect Ca^2+^ against direct contact with OH^−^ and subsequently prevented the formation of precipitate [16]. This result coincided with the effect of pH on MBP-Ca preparation (Figure 1B). Similarly, an investigation on the bioaccessibility of iron-chelating peptides extracted from collagen of tilapia skin showed that the bioaccessibility of both ferrous sulfate and iron chelates remained above 80% at pH = 1; within the pH range of 3–9, the bioaccessibility of ferrous sulfate dramatically decreased to 20.57%, while the that of iron chelates remained above 55% [54].

#### 3.3.2. Calcium Release Rates under Different Temperature Conditions

Thermal processing is a common food processing method that can change the sensory properties and the preservation period of food [55]. Therefore, investigating the thermal stability of MBP-Ca is of great significance for its application in food processing. Within the temperature range between 25–100 °C, the calcium release rates of MBP-Ca at different temperatures were little changed, all of which were above 90%; whereas, CaCl_2_ showed a calcium release rate lower than 70% at each temperature range, which was much lower than that of MBP-Ca within the same temperature range (Figure 4B). A similar phenomenon was observed by Zhang et al. [48] in the study of the stability of cattle bone collagen peptides-calcium chelates. While the temperature rised from 50 °C to 80 °C, there was no apparent variation in the calcium level in the chelate supernatant [48]. The aforementioned results indicated that MBP-Ca had good thermal stability and its calcium release rate was less affected by temperature.

#### 3.3.3. Calcium Release Rates during In Vitro Simulated Gastrointestinal Digestion

Bioavailability refers to the proportion of final circulating substances in the human body (the substances enter the blood or lymphatic tissues after absorption through the digestive tract) in total intake [43,56]. Bioaccessibility refers to the ratio of the substances released into gastrointestinal fluids to the total intake during gastrointestinal digestion, which represents the relative amount of the substances that can be absorbed by the human body, namely, the maximum absorbable amount [56,57]. Thus, the bioaccessibility analysis can provide an important basis for the assessment of bioavailability. Oral administration of dietary calcium supplements will largely affect the calcium release rate/bioaccessibility owing to exposure to multiple gastrointestinal digestive enzymes. Therefore, we investigated the calcium release rates of MBP-Ca and the positive control CaCl_2_ during gastrointestinal digestion. As shown in Figure 4C, the calcium release rate of MBP-Ca was significantly higher than that of CaCl_2_ under the simulated gastric digestion environment (0–120 min) at pH = 2 (*p* < 0.05); the calcium release rate of MBP-Ca remained at 80–95% while that of CaCl_2_ was maintained within the range of 50–70%. Subsequently, under the simulated intestinal digestion environment (pH = 7.4, 120–240 min), the calcium release rate of MBP-Ca sharply decreased to 17.51% and remained stable until the end of gastrointestinal digestion (240 min) (*p* > 0.05). In sharp contrast, the calcium release rate of CaCl_2_ decreased to 10.07% after treatment with simulated intestinal digestive fluid and finally dropped to 6.66% (*p* < 0.05). MBP-Ca exhibited a significantly higher calcium release rate than CaCl_2_ throughout the gastrointestinal digestion process (*p* < 0.05). MBP-Ca was insensitive to pepsin and an acidic environment and remained more stable. Although the presence of trypsin led to the degradation of MBP-Ca, the obtained small peptides could form new peptides-calcium chelate under an alkaline environment and preserve the protective effect on Ca^2+^ against the formation of Ca(OH)_2_ precipitate [58]. Similarly, Wang et al. [12] and Zhang et al. [51] showed similar findings on the bioaccessibility of calcium-binding peptides. Altogether, with the potential of improving the bioavailability of calcium, MBP-Ca held the promise as a highly efficient raw material for calcium supplements in the future.

## 4. Conclusions

In the current study, MBP was applied for the preparation of MBP-Ca, a new type of dietary calcium complement. The preparation process was optimized by orthogonal tests. Under optimal preparation conditions (pH = 6, temperature = 45 °C, time = 60 min, mass ratio of MBP/CaCl_2_ = 4:1, MBP concentration = 20 mg/mL), MBP-Ca could be efficiently synthesized with a calcium chelating rate up to 86.26%. The results of fluorescence spectra suggested that MBP-Ca was a new compound different from MBP. Amino acid composition and infrared spectroscopy analyses showed that oxygen atoms and nitrogen atoms of carboxyl, amino, and peptide bonds in MBP were the main chelation sites. The results of CD spectroscopy, SEM, and particle size distribution analysis revealed that the intra- and intermolecular interactions involving Ca^2+^ caused significant changes in the spatial structure of MBP and subsequent folding and aggregation of the peptides, which contributed to a more stable and orderly secondary structure, ultimately generating unique morphology of MBP-Ca. Most notably, compared with the conventional inorganic calcium salt CaCl_2_, MBP-Ca exhibited a superior calcium release rate under different temperatures, pH values, and in vitro simulated gastrointestinal digestion conditions. This was essential for subsequent calcium transport and absorption. Taken together, these findings suggested that MBP-Ca derived from mung bean was a potential dietary nutrient for improving calcium bioavailability. The in vivo absorption efficiency of MBP-Ca and its specific absorption mechanism warrant further investigation.

## Figures and Tables

**Figure 1 foods-12-01058-f001:**
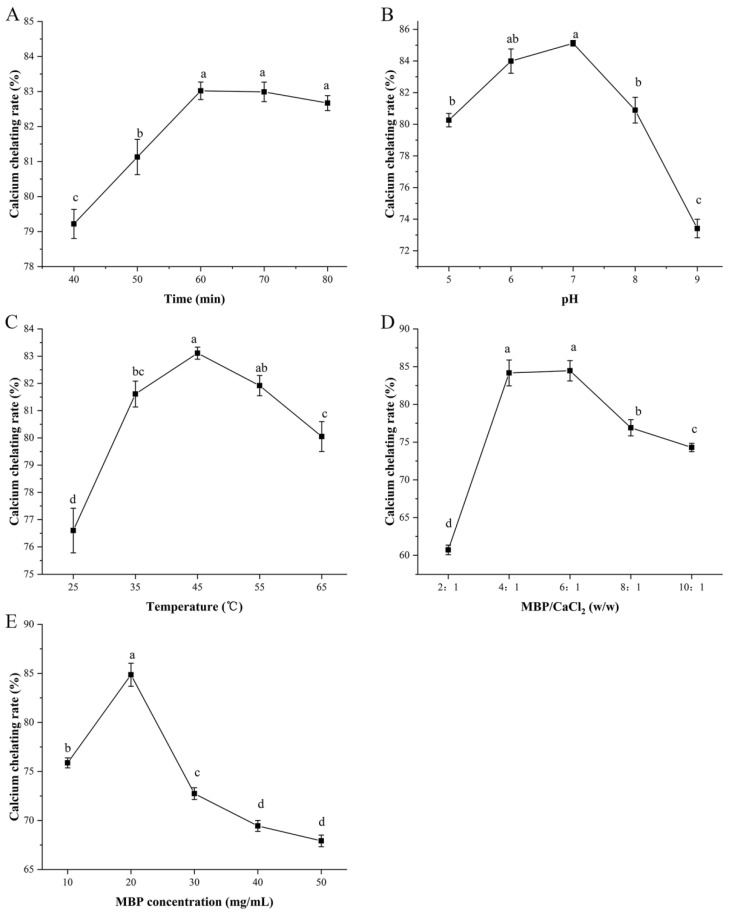
Effects of different process conditions on calcium chelating rate of MBP-Ca: (**A**) time, (**B**) pH, (**C**) temperature, (**D**) mass ratio of MBP/CaCl_2_, and (**E**) MBP concentration. Different letters indicate a significant difference (*p* < 0.05).

**Figure 2 foods-12-01058-f002:**
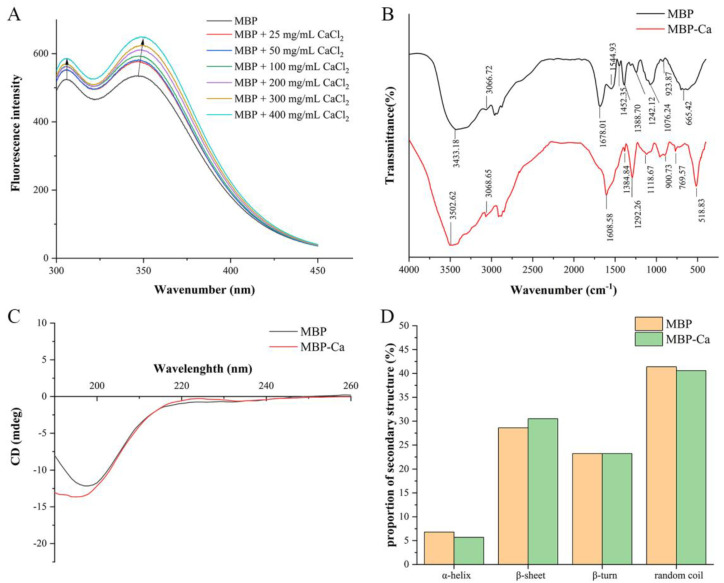
Structural characterization of MBP and MBP-Ca. (**A**) Fluorescence spectra of MBP with different concentrations of CaCl_2_, (**B**) FTIR spectra, (**C**) CD spectra, (**D**) The proportions of secondary structures of MBP and MBP-Ca calculated by CDNN 2.1 software.

**Figure 3 foods-12-01058-f003:**
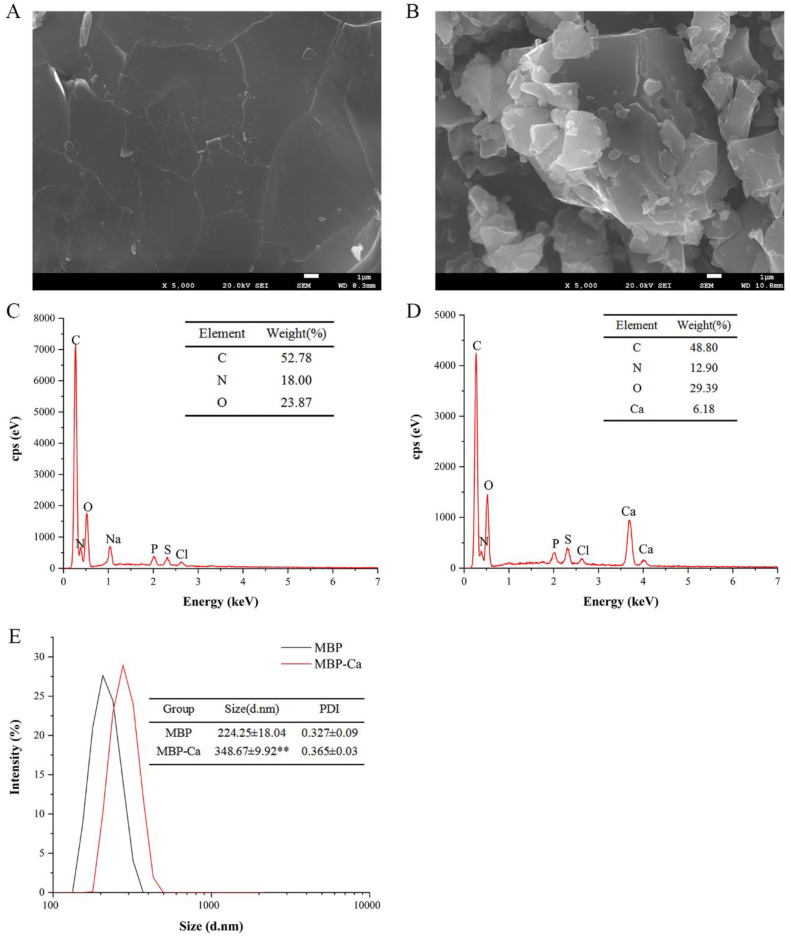
Morphological characterization of MBP and MBP-Ca. Scanning electron microscopic images of MBP (**A**) and MBP-Ca (**B**). Surface elemental compositions of MBP (**C**) and MBP-Ca (**D**). Particle size distribution of MBP and MBP-Ca (**E**). Asterisk indicates statistical significance (** *p* < 0.01) compared to the MBP control group.

**Figure 4 foods-12-01058-f004:**
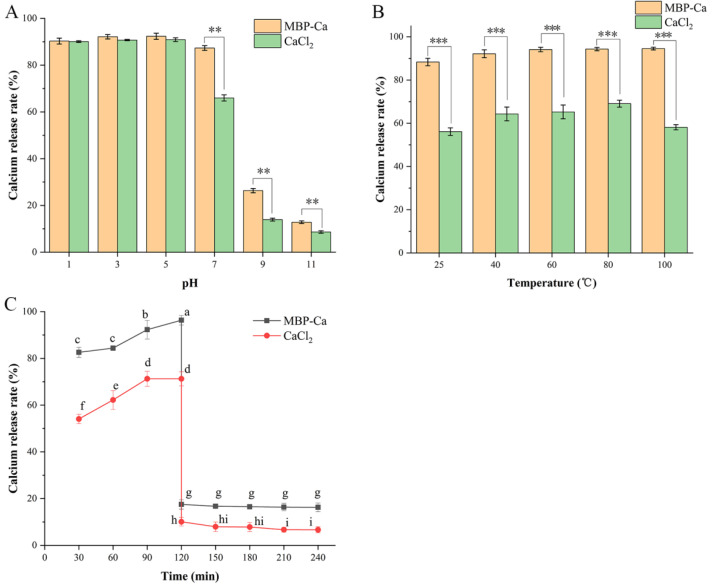
Calcium release rates of MBP-Ca and CaCl_2_ under different pH conditions (**A**), temperatures (**B**), and simulated gastrointestinal digestion in vitro conditions (**C**). ** *p* < 0.01 or *** *p* < 0.001, as compared to CaCl_2_ determined by Student’s *t* test in (**A**,**B**). Different letters indicate a significant difference at *p* < 0.05 determined by Duncan’s test in (**C**).

**Table 1 foods-12-01058-t001:** Orthogonal experimental design and results for the optimization of MBP-Ca preparation process.

Group Number	pH	Temperature (°C)	Mass Ratio of MBP/CaCl_2_ (*w*/*w*)	MBP Concentration (mg/mL)	Calcium Chelating Rate (%)
1	6	35	4:1	10	84.17 ± 0.60
2	6	45	6:1	20	85.16 ± 0.65
3	6	55	8:1	30	82.61 ± 0.44
4	7	35	6:1	30	83.60 ± 0.37
5	7	45	8:1	10	83.06 ± 0.80
6	7	55	4:1	20	85.66 ± 0.83
7	8	35	8:1	20	81.45 ± 0.30
8	8	45	4:1	30	83.29 ± 0.42
9	8	55	6:1	10	82.36 ± 0.95
k1	83.980	83.007	84.370	83.197	
k2	83.110	83.833	83.710	84.090	
k3	82.363	83.543	82.373	83.167	
R	1.747	0.756	1.997	0.923	

**Table 2 foods-12-01058-t002:** Amino acid compositions of MBP and MBP-Ca.

Amino Acids	MBP (%)	MBP-Ca (%)
Asp	12.14	15.10
Thr	2.91	2.29
Ser	4.61	4.87
Glu	21.31	32.74
Pro	4.79	3.92
Gly	3.43	3.55
Ala	4.12	3.37
Cys	0.13	0.19
Val	5.79	4.20
Met	0.29	0.08
Ile	4.90	3.86
Leu	8.66	4.19
Tyr	3.46	2.25
Phe	6.57	3.86
Lys	7.00	6.23
His	2.78	2.17
Arg	7.11	7.14
Acidic amino acids ^a^	33.45	47.84
Basic amino acids ^b^	16.89	15.54

^a^ containing Glu and Asp. ^b^ containing Lys, Arg, and His.

## Data Availability

Not applicable.

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
