# Peer review of "Process Optimization, Structural Characterization, and Calcium Release Rate Evaluation of Mung Bean Peptides-Calcium Chelate"

_foods, 2023, doi:10.3390/foods12051058_

Round 1

Reviewer 1 Report

The manuscript presents a successful attempt to develop a novel and efficient calcium supplement, using mung bean peptides (MBP) and CaCl2 as raw materials, and optimal conditions such as pH and temperature were proposed.  To achieve this, the scientists have conducted a broad spectrum of analysis. As a whole, the quality of the study and the paper is high.The Introduction provides the necessary information with relevant references cited. At the end the objectives of the study have been outlined, however, in my opinion, they need reforming. What is presented is more an achievement of an objective, rather than an objective. For instance: As formulated by the authors the first objective is: An orthogonal test design was applied to determine optimal preparation conditions of MBP-Ca. In my opinion an objective would rather sound like: 1. To determine the optimal preparation conditions of MBP-Ca through orthogonal design......

2. The structure and composition of MBP-Ca were characterized using multiple modern instrumental analysis techniques to be reformed as To characterise the structure and composition of MBP-Ca. (No need to point that you will use multiple modern techniques, this will revealed further in Material and methods section).

3) The calcium release rate of MBP-Ca was measured under varying temperatures, pH, and simulated gastrointestinal digestion conditions. To be reformed as: To measure the calcium releease rate of MBP-Ca under varying temperatures, pH, and simulated gastrointestinal digestion conditions.

The last sentence of the Introduction is not necessary. It is summary of the results that should be placed either in the Abstract or in the Conclusions.

The material and methods are described in details. Statistical evaluation however is missing the Student's test, that has been applied and pointed in Fig. 3 E and Fig 4, A and B. What is the reason to select Duncan instead of Student to compare the results in Fig. 4 C? In my opinion, T-test can be also applied since you compare the MBP-Ca nd CaCl2 as in the Fig 4 A and B. The only difference is that the factor is temperature.

The conclusions are sound and derived from the results.

Reviewer 2 Report

The manuscript is well written and can be  minor revisions as

1.Why mung bean peptides were selected? mention a little about importance of mung bean in abstract

2.About CD curve please mention the wavelengths in nm in R & D section

3. N-H stretching in FTIR spectra are close to O-H stretching Provide more evidences for confirming N-H stretching

4. Once check the style of references as per journal's guidelines 

Reviewer 3 Report

  Comments are on the manuscript copy download and respond carefully.

Similarity of the manuscript is 30% which is higher and two sources exceeding 5% so this is serious concern for the manuscript..

Reviewer 4 Report

Dear Authors,

this work titled "Process Optimization, Structural Characterization, and Calcium Release Rate Evaluation of Mung Bean Peptides-Calcium chelate is interesting.

The experimental plan seems supporting the claims and conclusions are sound and justified.

I recommend authors to revise the English for english grammar and spell check: presence of numerous errors in the manuscript.

Minor errors: 

Abstract:To (in first line) eliminate character bold 

References: change reference in references; reverse name and first name of author; change title, year and number of article in italics.   
